# Knowledge, Attitudes and Behaviours of Adolescents and Young Adult Population on the Use of E-Cigarettes or Personal Vaporizer

**DOI:** 10.3390/healthcare11030382

**Published:** 2023-01-29

**Authors:** Eduardo Sánchez-Sánchez, Lucía García-Ferrer, Guillermo Ramirez-Vargas, Jara Díaz-Jimenez, Manuel Rosety-Rodriguez, Antonio Jesús Díaz, Francisco Javier Ordonez, Miguel Ángel Rosety, Ignacio Rosety, Miriam Poza-Méndez

**Affiliations:** 1Internal Medicine Department, Punta de Europa Hospital, 11207 Algeciras, Spain; 2Department of Nursing and Physiotherapy, University of Cadiz, 11009 Cadiz, Spain; 3Biomedical Research and Innovation Institute of Cadiz, Puerta del Mar University Hospital, University of Cadiz, 11009 Cadiz, Spain; 4San Juan Grande Hospital, 11408 Jerez de la Frontera, Spain; 5Medicine Department, School of Medicine, University of Cádiz, Fragela Square s/n, 11003 Cadiz, Spain; 6Human Anatomy, School of Medicine, University of Cádiz, Fragela Squares/n, 11003 Cadiz, Spain; 7Move-It Research Group, Biomedical Research and Innovation Institute of Cadiz, Puerta del Mar University Hospital, University of Cádiz, Fragela Squares/n, 11003 Cadiz, Spain

**Keywords:** adolescent, e-cigarette vapor, personal vaporizer, dangerous behaviour, vape, vaping, adult

## Abstract

The use of electronic cigarettes (e-cig) or personal vaporizers (PV) has increased in recent years, especially among adolescents and adults, increasing risk factors for their health or being a starting point for other risk behaviours. The aim of our study is to learn the knowledge, attitudes and behaviours of the adolescent and young adult population on the use of e-cigarettes or PVs. A cross-sectional descriptive study was carried out among 375 subjects. The use of e-cigs or PVs was measured using a self-administered questionnaire. Overall, 13.33% of the respondents used e-cigarettes or PVs. A correlation could be established between vape use and age, with 14–18-year-olds being the highest users (25.9%). Additionally, 90.13% of the respondents answered that e-cig or PV use was harmful to health. Other behaviours, such as consumption of energy drinks (ED), increases the probability of vaping by 3.08 times (CI = 1.55–6.29; *p* = 0.001). Subjects aged 23–26 years and 27–34 years are less likely to vape than subjects aged 14–18 years (OR = 0.31: CI = 0.09–0.96; *p* = 0.044; OR = 0.07: CI = 0.00–0.63; *p* = 0.037). The same applies to subjects with secondary education (OR = 0.17: CI = 0.04–0.66; *p* = 0.011) and postgraduate education (OR = 0.07: CI = 0.06–1.19; *p* = 0.043), than subjects with primary education. The use of e-cigs and PVs starts at an early age, despite the known harmful effects of e-cigs and PVs. Their use may occur mainly in recreational situations and in association with other substances such as ED.

## 1. Introduction

During the adolescence period or among adults, risk behaviours are common, such as using harmful substances [1]. The type of substance consumed has been developing in the course of time, and new devices for its consumption have appeared. Within these substances and devices, we can find the electronic cigarette (e-cig) or personal vaporizer (PV). These electronic devices are designed for the inhalation of a liquid converted into an aerosol. There are different types of liquids with sweet, fruit and mint flavours, sometimes also containing nicotine [2].

Use of e-cigs and PVs has increased in the last few years among the adolescent and adult population [3,4]. E-cigs and PVs appeared as a “healthier” alternative to tobacco, as a first step to giving up smoking [5,6] and to avoid new anti-smoking laws. These devices are capable of vaporizing a nicotine or nicotine-free solution combined with flavoured liquids [7]. The use of these devices may potentially play a role in social imbalance, learning and academic performance in this population [6].

Furthermore, some authors have concluded that e-cig consumption increases health threats in this population in both t short-and long-term [8]. In 2018, the increase in e-cig and PV use among young people was considered an epidemic in the USA. In 2019, an emerging disease gained awareness in USA, where patients had serious lung injuries associated with the use of e-cigs and PVs (EVALI) [9,10]. Additionally, according to the reported data from Liu et al. in 2020, smokers have a higher probability of disease progression of COVID-19 [11].

E-cig and PV use is associated with other behaviours or factors such as age, socio-economic status, physical activity or the idea that they are healthier or less harmful than other behaviours [12]. In addition, male gender, stress and everyday situations such as studies or work may lead to an increase in e-cig or PV use among adolescents [13,14].

Other studies show that those individuals who use e-cigs or PVs are more likely to use other substances [15], including energy drinks. Some studies have shown a strong association between e-cig and PV use and energy drink consumption [16,17]. This relationship could also be the other way around, as Fagar et al. in 2020 concluded that energy drink consumption increased the likelihood of using an e-cigs and PVs by 4.66 times [18].

In the past, sports people were the major consumers of energy drinks (ED), with the aim to reduce fatigue associated with exercise [19], but due to their normalization and the increasing variety existing on the market [20], the consumption of these drinks has become very popular among adolescents and adults, presenting an important increase in the last few years [21,22]. The majority of the consumption of these drinks is linked to smoking habits and alcohol, as well as risky sexual behaviour and a higher accident rate [23].

No published articles have been found that relate e-cig and PV use to behavioural or daily life behaviours, such as exercise, school/work performance, sleeping hours, night-time leisure or other newly emerging behaviours such as the consumption of energy drinks. It is, therefore, necessary not only to know the prevalence of e-cigarette or PV use, but also how these can affect the daily life of the population that consumes them and the circumstances in which this consumption takes place, with the aim of creating preventive health policies to reduce or delay their use and, thus, avoid the appearance of other risk behaviours or health problems in the adolescent and young adult population.

The aim of our study is to learn the knowledge, attitudes and behaviours of the adolescent and young adult population on the use of e-cigarettes or personal vaporizers.

## 2. Materials and Methods

### 2.1. Study Design and Participants

A cross-sectional descriptive study. Non-probability convenience and snowball sampling were used, where the participants themselves disseminate and recruit other participants.

The inclusion criteria for the sample were that the participants had to be between 14 and 34 years of age and who voluntarily want to participate.

### 2.2. Instruments and Variables

The sociodemographic variables collected were gender, age, residence, socioeconomic status, study level, and employment status.

A standardized questionnaire about vaping in the adolescent and adult population was not found. For that reason, a structured questionnaire prepared by the authors was designed, taking as reference other questionnaires about tobacco and drugs consumption. Advice from a focus group formed by 5 members was sought, and some questions were modified due to the duality of its contents.

The questionnaire for data collection was divided into 3 sections. In the first section, sociodemographic variables were collected; in section two, behavioural and life variables were collected; and in the last section, variables related to vaping were collected. The variables and questions are shown in the different tables in the results section.

### 2.3. Data Collection

The questionnaire was administered online via a free platform (Google Forms) (https://drive.google.com/file/d/1ocGdcdIYtJpqcHw8RZCvlK4YclCKXoTX/view?usp=share_link (accessed on 12 December 2022)) [24]. Social networks such as WhatsApp, Twitter, Facebook, and Instagram were used for its dissemination. The questionnaire was administered between March and April of 2020.Completion of the questionnaire was voluntary.

### 2.4. Statistical Analysis

The statistical treatment was carried out with IBM SPSS Statistic 25 software (IBM SPSS Statistics for Windows, Version 25.0, IBM Corp., Armonk, NY, USA). A descriptive analysis was carried out to determine the sample distribution for each of the variables studied. The qualitative variables were represented by frequency and percentage, and the quantitative variables were expressed by the mean and standard deviation or dispersion.

Subsequently, and using the X^2^ test, it was studied whether there were significant differences between age and the use of e-cigs or PVs. In addition, the relationship between vaping and energy drinks was studied, as well as the relationship between vaping and the independent variables studied.

A linear regression model was performed within the independent variables: age, gender, place of residence, work and consumption of energy drinks, for the variable of vaping use, obtaining OR values.

Statistical significance was determined for a value of *p* < 0.05.

### 2.5. Ethical Considerations

This work was conducted in accordance with the Declaration of Helsinki. The aim of the study and the anonymity of participants, as well as the voluntary character of participation, were all explained before the participants started answering the questionnaire and their informed consent was obtained. The participants were also informed that the data obtained would be used for research purposes only.

## 3. Results

The questionnaire was answered by a total of 386 subjects. Eleven questionnaires were eliminated in total, two because they did not answer all the items, and nine contradicted their answers. Ultimately, the sample was 375 subjects.

### 3.1. Socio-Demographic Variables

The sample was 375 participants: 71.73% (*n* = 269) were females, and 28.27% (*n* = 106) were males. Subjects between 19- and 22-years-old were the most prevalent (43.20%, *n* = 162). Regarding the place of residence, 78.67% (*n* = 295) lived at the family home, against 2.13% (*n* = 8) that lived independently and 19.20% (*n* = 72) in shared accommodation. According to socioeconomic status, 81.65% (*n* = 307) were classified as middle level, followed by 16.53% (*n* = 62) of low level. Overall, 53.60% of the surveyed were studying or had already studied university degrees, and 0.27% did not possess any educational qualifications. On the other hand, 64.53% of the surveyed were studying, 17.87% worked and 17.60% were doing both things (Table 1).

### 3.2. Vaping Consumption and Health

Overall, 25.9% of the subjects aged 14–18 years were vaping, with statistically significant differences in the different age groups. Additionally, 90.13% of the respondents answered that e-cig or PV use was harmful to their health (*p* = 0.020), and 52.80% answered that they were not healthier than smoking (Table 2).

### 3.3. Behaviours and Characteristics of Vapers

Among the subjects who vaped, 40.0% (*n* = 50) used an electronic vaporizer (EV), the subjects from 19- to 22-years-old being the ones who use it the most (58.8%). Additionally, 34.0% of respondents did not know or distinguish the type of device used.

The nicotine-free and flavoured substance was the most used (62.0%). The subjects who used the most flavouring and nicotine were between 19- and 22-years-old. Fruit flavours were the most commonly used (42.0%), being most used especially by subjects aged 14–18 years and 23–26 years, with 33.3%, respectively.

Overall, 75% of the subjects using vaping as a tobacco substitute were those aged 23–26 years. The most common answers by the subjects of the study and that reflect this figure are: “because I like the flavours”, “simply I like it”, “it distracts me” and “it relaxes me”. In total, 68.0% vaped when with friends, and 100.0% vaped in all the listed situations. These last two variables were statistically significant among the different age groups (Table 3).

### 3.4. Variables Correlated with Vaping

Energy drinks consumption was more prevalent in participants who vaped (68.0%, *p* < 0.001). We assessed whether using an e-cig or PV influenced different physical or behavioural variables such as health perception, physical exercise, sleep, nightlife, academic/job performance, and difficulty with social interaction. The results showed that vapers had a more active nightlife (*p* = 0.006). Although the other variables did not indicate statistically significant differences (*p* > 0.05), academic/job performance was weaker in participants who vaped (Table 4).

A generalised linear regression model is performed, first taking vaping consumption as the dependent variable. The results obtained are shown in Table 5. In this table, we can see that subjects who consume ED are 3.08 times more likely to vape than those who do not (CI = 1.55–6.29; *p* = 0.001). Subjects aged 23–26 years (OR = 0.31: CI = 0.09–0.96; *p* = 0.044) and 27–34 years (OR = 0.07: CI = 0.00–0.63; *p* = 0.037) are less likely to vape than subjects aged 14–18 years. The same is true for subjects with secondary education (OR = 0.17: CI = 0.04–0.66; *p* = 0.011) and postgraduate degree (OR = 0.07: CI = 0.06–1.19; *p* = 0.043), than subjects with primary education.

## 4. Discussion

According to reported data from some government agencies and scientific studies, the consumption of vaping has increased between adolescences and adult population in the last years, and it is predicted that this figure will raise [25,26]. There are many authors that have focused their researches on knowing the prevalence of vaping and the factors that increase the likelihood of vaping.

In an article published by Demissie et al. in 2017, it was observed that 15.8% of secondary students used personal vaporizers and 7.5% used personal vaporizers and traditional cigarettes [3]. These data are higher than the data obtained in our sample (13.33%), although if we take only the group of comparable age to secondary students, 14–18-years-old subjects, as reference, the prevalence increases up to 25.9%. In the study carried out by Okawa et al. in 2014 with participants between 15- and 29-years-old, the results were very much lower, the prevalence being 4.3% [27]. Another remarkable aspect is the model of devices for vaping. The personal vaporizer is the most used in our sample; however, it highlights that a great percentage (34.0%) do not know which type of device they are using or the differences between all devices.

While there is an existing correlation between using e-cigs and PVs and a smoking history, an important number of adolescents and adults that used these devices had never smoked [28]. E-cigarettes and personal vaporizer users felt that its consumption provides more benefits than traditional cigarettes, such as: having more friends, feeling well, feeling more comfortable in social situations, etc. [29]. Consumption for pleasure, for fun or as a relaxing method are the most prevalent in the studied population, especially in subjects under the age of 26 years. Moreover, this consumption was more frequent when they were meeting friends, this percentage being higher in adolescents within the age range of 19–22 years (41.42%). This search for pleasure, fun and to relax is performed above the idea that using vaporizers is harmful for their health (90.13%), but their consumption was healthier than traditional cigarettes (52.80%). Some studies have found a relation between the use of e-cigarettes, especially those with nicotine, and the use of traditional cigarettes and marihuana [30,31].

Perhaps this search for pleasure and fun may be related to a broad range of flavours and the idea of being healthier than cigarettes [32]. According to data reported by Soneji et al. in 2019, fruity, sweet and menthol flavours were the most common flavours used by adolescents (12–17 years) and young adults (18–24 years), against the use of nicotine and other flavours that were used by older adults (≥25 years) [33]. These figures are very different to those obtained in our study, as, although 62.0% of our subjects consumed only a flavour, 38.0% consumed flavour and nicotine, this percentage being higher in subjects between 27–30-years-old (100.0%) and between 14–18 years (42.8%). Within the flavours, the majority of our subjects used fruity flavours. Perhaps the range and kinds of flavours promote the increase of the prevalence of consumption in this population.

Different studies have obtained very variable results about the use of these substances and different parameters, such as academic performance, sleep habits, perceived health condition, etc. [6]. In our study, the number of subjects that vaped was greater in those with a more frequent nightlife, this conclusion being similar to other studies [17].

No differences in physical activity, rest and sleep hours were found between subjects who vaped and those who did not. This result is different from that reported by other authors [12]. Differences were found in night-time recreation, with those who vaped reporting a higher number of days of night-time recreation. Subjects who used e-cigarettes or personal vaporizers reported poorer school/work performance and more difficulty concentrating on some occasions.

There are some authors that have reported that male sex, medium socioeconomic level, type of diet, etc., increase the probability of consuming vaporizers [30]. In our study, socio-economic status, household residence, gender and occupation (study or work) influenced the likelihood of vaping. There were differences related to age and socio-economic status. Subjects aged 14–18 years were more likely to vape than those aged 23–34 years. Other authors have obtained the same results, i.e., younger subjects are more likely to initiate e-cig and PV use. In addition, the consumption of energy drinks increases the likelihood of e-cig and PV use. In the study carried out in Finland by Kinnunen in 2018, they concluded that the consumption of energy drinks was a strong predictor of e-cigarettes use [17]. This result is similar to the one obtained in our study, where it was found that an association between subjects that consumed e-cigs and personal vaporizers and energy drinks [16].

### Limitationsand Strengths

Among the limitations of the study, the type of sampling used in this study may give rise to a selection bias, as the sample was obtained non-randomly. This bias has been assumed due to the difficulty of accessing the entire Spanish population. Access to the study population at the national level was very difficult due to the situation in which the Spanish health system was immersed, and the mobility restrictions of the time (COVID-19 lockdown from 15 March). These restrictions may have caused a bias in the responses, as this situation may have led to changes in e-cigarette consumption, as published by Galus et al., although the responses were related to pre-lockdown behaviours [34].

The under-representation of men relative to the population could lead to a gender bias. Another limitation present is the difference in representation by age, as the profile of use and risk behaviours may be modified by age, so results should be interpreted with caution. In addition, the use of an online questionnaire may lead to a response bias due to acquiescence, although the questionnaire was simplified to reduce the response time. However, Ekman et al. in 2006 stated that the bias with the collection of information through web-based questionnaires was no greater than that caused by paper-based questionnaires [35]. Another limitation may be the use of an unvalidated questionnaire.

The generalisability of the results is limited due to the non-probability sample and the biases mentioned above. However, it should be noted that, although the present research cannot be representative of the entire Spanish population, an adequate sample has been achieved, being a starting point for future more specific research.

Despite the limitations encountered, our study has strengths to be considered. It is one of the first studies in valuing the consumption of e-cigarettes or personal vaporizers in adolescent and young adult populations in Spain, providing relevant data on the knowledge, attitudes and behaviours of this population regarding the use of e-cigarettes or personal vaporizers. Given the difficulty of data collection due to restrictions, a sizeable sample was drawn to overcome the shortcomings of the non-probabilistic sampling method of convenience.

## 5. Conclusions

The use of e-cigs and PVs starts at an early age, despite their known harmful effects. This use may occur mainly in recreational situations and in association with other substances such as energy drinks. Its use can lead to inappropriate behaviour, especially when consumed with other substances.

These data suggest that not only awareness or education campaigns should be carried out, but also other governmental measures to prevent their use, such as restricting their sale at an early age, increasing their prices, etc., as the use of e-cigarettes and PV can be a global public health problem.

Further research is needed on modifying preventive behaviours to reduce or delay their use, their relationship with other risk behaviours, and the medium- and long-term health problems associated with their use.

## Figures and Tables

**Table 1 healthcare-11-00382-t001:** Socio-demographic variables.

	n	%
**Gender**		
Female	269	71.73
Male	106	28.27
**Age range**		
14–18 years	54	14.40
19–22 years	162	43.20
23–26 years	124	33.07
27–34 years	35	9.33
**Residence**		
Family	295	78.67
Shared accommodation	72	19.20
Alone	8	2.13
**Socioeconomic status**		
Low	62	16.53
Medium	307	81.87
High	6	1.60
**Study level**		
No formal education	1	0.27
Primary education (School)	13	3.47
Secondary education (College/High School)	126	33.60
University degree	201	53.60
Postgraduate degree	34	9.07
**Currently, what do you do?**		
Studying	242	64.53
Working	67	17.87
Studying and working	66	17.60

**Table 2 healthcare-11-00382-t002:** Vaping consumption and health.

Questions		Age Range (n, %)				
	14–18	19–22	23–26	27–34	Total	*p*
**Do you vape?**						0.010 *
Yes	14 (25.9)	21 (13.0)	14 (11.3)	1 (2.9)	50 (13.33)	
No	40 (74.1)	141 (87.0)	110 (88.7)	34 (97.1)	325 (86.67)	
**Do you think that vaping is harmful?**						0.020 *
Yes	42 (77.8)	151 (93.2)	114 (91.9)	31 (88.6)	338 (90.13)	
No	2 (3.7)	3 (1.9)	4 (3.2)	-	9 (2.40)	
Do not know	10 (18.5)	8 (4.9)	6 (4.8)	4 (11.4)	28 (7.47)	
**Do you think that vaping is healthier than traditional cigarettes?**						0.552
Yes	20 (37.0)	55 (34.0)	30 (24.2)	10 (28.6)	115 (30.67)	
No	25 (46.3)	80 (49.4)	73 (58.9)	20 (57.1)	198 (52.80)	
Do not know	9 (16.7)	27 (16.7)	21 (16.9)	5 (14.3)	62 (16.63)	

Note: * *p* < 0.05.

**Table 3 healthcare-11-00382-t003:** Behaviours and characteristics of vapers.

Questions	Age Range (n, %)					
	14–18	19–22	23–26	27–34	Total	*p*
**Which type of vaporizer do you use?**						0.602
E-cig	0 (0.0)	1 (100.0)	0 (0.0)	0 (0.0)	1 (2.0)	
EV	4 (23.5)	10 (58.8)	3 (17.6)	1 (5.0)	20 (40.0)	
MV	6 (30.0)	7 (35.0)	6 (30.0)	0 (0.0)	9 (18.0)	
SMV	4 (44.4)	1 (11.1)	4 (44.4)	0 (0.0)	3 (16.0)	
Do not know	0 (0.0)	2 (66.7)	1 (33.3)	0 (0.0)	17 (34.0)	
**Which substance do you use when you vape?**						0.573
Nicotine	0 (0.0)	0 (0.0)	0 (0.0)	0 (0.0)	0 (0.0)	
Nicotine-free and flavoured substance	8 (25.8)	14 (45.2)	9 (29.0)	0 (0.0)	31 (62.0)	
Nicotine and flavours	6 (31.0)	7 (36.8)	5 (26.3)	1 (5.3)	19 (38.0)	
**Which flavour do you use?**						0.806
Sweet	1 (25.0)	2 (50.0)	1 (25.0)	0 (0.0)	4 (8.0)	
Fruit	7 (33.3)	6 (28.6)	7 (33.3)	1 (4.8)	21 (42.0)	
Mint	3 (30.0)	6 (60.0)	1 (10.0)	0 (0.0)	10 (20.0)	
All of them	3 (20.0)	7 (46.1)	5 (33.3)	0 (0.0)	15 (30.0)	
**Why do you use vaporizers?**						0.009 **
For fun	5 (29.4)	9 (52.0)	3 (17.6)	0 (0.0)	17 (34.0)	
For pleasure	5 (27.8)	6 (33.3)	7 (38.9)	0 (0.0)	18 (36.0)	
To relax myself	4 (36.4)	6 (54.5)	1 (9.1)	0 (0.0)	11 (22.0)	
As substitute of traditional cigarettes	0 (0.0)	0 (0.0)	3 (75.0)	1 (25.0)	4 (8.0)	
**In which circumstances do you vape?**						0.003 **
With friends	12 (35.3)	14 (41.2)	8 (23.5)	0 (0.0)	34 (68.0)	
With family	1 (50.0)	1 (50.0)	0 (0.0)	0 (0.0)	2 (4.0)	
Party time	0 (0.0)	1 (50.0)	1 (50.0)	0 (0.0)	2 (4.0)	
At home	0 (0.0)	1 (50.0)	0 (0.0)	1 (50.0)	2 (4.0)	
Alone and friends	1 (12.5)	4 (50.0)	3 (37.5)	0 (0.0)	8 (16.0)	
All of them	0 (0.0)	1 (50.0)	2 (100.0)	0 (0.0)	2 (4.0)	

E-cig: electronic cigarette; EV: electric vaporizer; MV: mechanical vaporizer; SMV: semi-mechanical vaporizer. Note: ** *p* < 0.005.

**Table 4 healthcare-11-00382-t004:** Relation between variables and vaping.

	Vaping		Total	*p*
	YES	NO		
**Do you use energy drinks?**				<0.001 ***
Yes	34 (68.0)	109 (33.5)	143 (38.13)	
No	16 (32.0)	216 (66.5)	232 (61.87)	
**What is your health like compared to others?**				0.330
Very good	12 (24.0)	59 (18.2)	71 (18.93)	
Good	31 (62.0)	203 (62.5)	234 (62.40)	
Regular	5 (10.0)	58 (17.8)	63 (16.80)	
Bad	2 (4.0)	4 (1.2)	6 (1.60)	
Very bad	0 (0.0)	1 (0.3)	1 (0.27)	
**How many hours do you sleep during night time?**				0.748
<4 h	0 (0.0)	2 (0.6)	2 (0.53)	
4–6 h	7 (14.0)	38 (11.7)	45 (12.00)	
6–8 h	31 (62.0)	222 (68.3)	253 (67.47)	
>8 h	12 (24.0)	63 (19.4)	75 (20.00)	
**Do you rest well?**				0.530
Yes	33 (66.0)	238 (73.2)	271 (72.27)	
A bit	15 (30.0)	74 (22.8)	89 (23.73)	
No	3 (4.0)	23 (4.0)	15 (4.0)	
**Do you practice physical exercising?**				0.680
Daily	2 (4.0)	22 (6.8)	24 (6.40)	
5/week	9 (18.0)	55 (16.9)	64 (17.07)	
3/week	21 (42.0)	106 (32.6)	127 (33.87)	
1/week	8 (16.0)	60 (18.5)	68 (18.13)	
Never	10 (20.0)	82 (25.2)	92 (24.53)	
**How is the intensity of the exercise (measured with speech-test)?**				0.598
Light (you can talk without problems)	14 (28.0)	65 (20.0)	79 (21.07)	
Moderate (you speak with difficulty, only short sentences)	23 (46.0)	161 (49.5)	184 (49.07)	
Intense (you can only pronounce monosyllables)	4 (8.0)	25 (7.7)	29 (7.73)	
Not applicable	9 (18.0)	74 (22.8)	83 (22.13)	
**How often do you go out on nightlife?**				0.006 *
>2/week	12 (24.0)	29 (8.9)	41 (10.93)	
2/week	14 (28.0)	66 (20.3)	80 (21.33)	
1/week	10 (20.0)	69 (21.2)	79 (21.07)	
1–3/month	11 (22.0)	92 (28.3)	103 (27.47)	
<1/month	2 (4.0)	51 (15.7)	53 (14.13)	
Never	1 (2.0)	18 (5.5)	19 (5.07)	
**Academic/job performance:**				0.120
Very good	11 (22.0)	114 (35.1)	125 (33.33)	
Good	30 (60.0)	81 (55.7)	211 (56.57)	
With difficulties	8 (16.0)	25 (7.7)	33 (8.80)	
Do not know	1 (2.0)	5 (1.5)	6 (1.60)	
**Do you have concentration difficulties?**				0.083
Yes	9 (18.0)	30 (9.2)	39 (10.40)	
Sometimes	24 (48.0)	201 (61.8)	225 (60.00)	
No	17 (34.0)	94 (28.9)	111 (29.6)	

Note: * *p* < 0.05; *** *p* < 0.001.

**Table 5 healthcare-11-00382-t005:** Odds ratio values (OR).

Variables	Vaping		
	OR	CI 95%	*p*
**Gender [Female]**			
Male	1.90	0.91–3.91	0.081
**Age range (years)** [14,15,16,17,18]			
19–22	0.46	0.16–1.27	0.131
23–26	0.31	0.09–0.96	0.044 *
27–34	0.07	0.00–0.63	0.037 *
**Residence [Familiar]**			
Shared accommodation	1.23	0.49–2.87	0.066
Alone	5.32	0.82–3.29	0.636
**Currently, what do you do? [Studying]**			
Studying and working	1.14	0.41–2.93	0.784
Working	1.51	0.50–4.32	0.446
**Socioeconomic level [High]**			
Low	0.49	0.05–1.06	0.558
Medium	0.49	0.06–1.00	0.541
**Study level [primary education]**			
No formal education	0.00	0.00–0.00	0.982
Secondary education	0.17	0.04–0.66	0.011 *
University degree	0.27	0.00–0.71	0.082
Postgraduate degree	0.07	0.06–1.19	0.043 *
**ED consumption [No]**			
Yes	3.08	1.55–6.29	0.001 **

Note: * *p* < 0.05; ** *p* < 0.005.

## Data Availability

The data are collected in a database prepared by the research team.

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
