# Peer review of "Knowledge, Attitudes and Behaviours of Adolescents and Young Adult Population on the Use of E-Cigarettes or Personal Vaporizer"

_healthcare, 2023, doi:10.3390/healthcare11030382_

Round 1

Reviewer 1 Report

The objective of the study is to know the prevalence of electronic cigarette consumption and to establish a profile of young consumers in Spain. It is a behavior that has been little studied and that it is necessary to know in order to establish adequate and effective prevention policies. The references of the related investigations are very novel and key to the justification of the study.

On the other hand, I believe that the study design poses some limitations as the authors themselves have described.

In relation to the participants, I think it was a mistake to mix adolescents and young people, since the consumption profile is not the same between the two age groups, nor is the same relationship established with other risky behaviors.

The results are clearly presented, but despite the fact that the discussion is well founded, I have not found the relationship with the objective of the investigation. They have not described the consumer profile.

I think they could have described the questionnaire developed more and on which they have been based. And having carried out another more specific analysis to elaborate the profile (ANOVA type).

Author Response

Dear reviewer,

Firstly, we are grateful for the time spent on our manuscript, as well as for the comments made that help us to improve the manuscript.

In the drafting of the project, we thought it would be convenient to carry out the questionnaire in adolescents and young people, as it would help us to know how consumption or associated behaviours evolve, but we understand that it could lead to a bias due to the consumption profile. This has been added in the limitations.

The aim has been modified so that it is more related to the statistical analysis carried out. We believe that the discussion is more related to the wording of the objective.

A link to the questionnaire used has been added to make it available to readers.

Once again, we thank you for your time and attention to our manuscript. We truly hope that we have met your expectations, with the modifications introduced.

Kind regards. 

Reviewer 2 Report

“Interesting and current work well written then can be published with the following reccomendations:

1.      Talk about the importance of prevention and information on new forms of tobacco, this topic should be included both in school and in the University in Medical courses in order to prepare future Medical Doctors (see the paper: Milella MS et al. E-learning course improves knowledge in tobacco dependence, electronic nicotine delivery systems and heat-not-burn products in Medical School students. Clin Ter. 2021 Sep 29;172 (5):427-434. doi: 10.7417/CT.2021.2353

2.      page 3, line 107-108, highlighted in yellow), insert: “The questionnaire should be made available upon request”. This is important for the readers.

3.      page 4 line 121, highlighted in yellow: “The questionnaire was administered between March and April 2020”. In this period Covid-19 lockdown has affected tobacco exposures in young people, discuss this issue (see the paper: Gallus S et al. Use of electronic cigarettes and heated tobacco products during the Covid-19 pandemic. Sci Rep. 2022 Jan 13;12(1):702. doi: 10.1038/s41598-021-04438)

Author Response

Dear reviewer,

Firstly, we appreciate the time dedicated to our manuscript, as well as the clarifications you request, which help us to understand the doubts that a future reader may have, if the manuscript gets published.

As you suggest it is important that prevention and information about new forms of tobacco should be included in the training of health professionals (doctors, nurses, ...), but we believe that it is more important to create social and health policies in order to provide this information to the whole population with the aim of empowering them to make decisions on this issue.

A link to the questionnaire used has been added to make it available to readers.

Although the data collection was at the beginning of the COVID-19 lockdown, so the responses were associated with previous behaviours, this point has been added in the limitations of the study, using the proposed article as a reference.

English language analysis has been redone by the authors and an English-speaking colleague.

Once again, we appreciate the time and attention dedicated to our manuscript. We really hope we have reached your expectations, with the modifications made and that the explanations to those that we have not modified be considered as appropriate.

Kind regards.

Reviewer 3 Report

Comment 01: Tittle is misleading :Use of electronic cigarettes and personal vaporizer in Adolescent and Young adult population. Title has to convey these :Online survey,  among young  adults on knowledge of e-cig or pv use or attitude towards e-cig or PV use along with energy drink.

Comment 02:statistica analysis part required improvement. 

Comment 03: Table 1. Socio-demographic variables- is a descriptive table. There is no need to do statistical test for difference between the categories

Comment 04:Number of Tables can be reduced.

Comments 05: response rate, selection bias, and subjective reporting are stated as limitations but not addressed sufficiently to validate the findings of the study

Author Response

Dear reviewer,

Firstly, we appreciate the time dedicated to our manuscript, as well as the clarifications you request, which help us to understand the doubts that a future reader may have, if the manuscript gets published.

As suggested, the title has been modified to better meet the aim of the study.

Although there is always room for improvement, we believe that the statistical analysis is adequate to meet the proposed objective. Please help us to know what improvement you are referring to.

Column p in table 1 has been removed as you suggested, as it is a descriptive table.

We believe that the use of tables may help the reader to understand the manuscript. 

The limitations section has been modified, as suggested by the reviewers. If you think we should add any new limitations, please help us. 

English language analysis has been redone by the authors and an English-speaking colleague.

Once again, we appreciate the time and attention dedicated to our manuscript. We really hope we have reached your expectations, with the modifications made and that the explanations to those that we have not modified be considered as appropriate.

Kind regards